# Safety and Efficacy of Incorporating Actellic^®^ 300 CS into Soil Wall Plaster for Control of Malaria Vectors in Rural Northeastern Uganda

**DOI:** 10.3390/tropicalmed10010004

**Published:** 2024-12-24

**Authors:** Tonny Jimmy Owalla, Emmanuel Okurut, Kenneth Ssaka, Gonsaga Apungia, Barbara Cemeri, Andrew Akileng, Basil Ojakol, Mark J. I. Paine, Hanafy M. Ismail, Thomas G. Egwang

**Affiliations:** 1Med Biotech Laboratories, Kampala P.O. Box 9364, Uganda; tjowalla@uw.edu (T.J.O.); okurutemmanuel1@gmail.com (E.O.); apungiagonsaga87@gmail.com (G.A.); barbaracemeri@gmail.com (B.C.); andrewaki94@gmail.com (A.A.); basiloj99@gmail.com (B.O.); 2Department of Global Health, University of Washington, Seattle, WA 98105, USA; 3School of Clinical Medicine and Dentistry, Western Campus, Kampala International University, Kampala P.O. Box 20000, Uganda; 4Department of Community Health, Kiryandongo General Hospital, Kiryandongo P.O. Box 351620, Uganda; kessaka@gmail.com; 5Department of Vector Biology, Liverpool School of Tropical Medicine, Liverpool L3 5QA, UK; mark.paine@lstmed.ac.uk (M.J.I.P.); hanafy.ismail@lstmed.ac.uk (H.M.I.)

**Keywords:** house decoration for malaria control, HD4MC, IRS, safety, efficacy: pirimiphos-methyl CS, Actellic^®^ 300 CS, organophosphate, malaria, vectors

## Abstract

Indoor residual spraying (IRS) and the use of insecticide-treated bednets for malaria vector control have contributed substantially to a reduction in malaria disease burden. However, these control tools have important shortcomings including being donor-dependent, expensive, and often failing because of insufficient uptake. We assessed the safety and efficacy of a user-friendly, locally tailored malaria vector control approach dubbed “Hut Decoration for Malaria Control” (HD4MC) based on the incorporation of a WHO-approved insecticide, Actellic^®^ 300 CS, into a customary hut decoration practice in rural Uganda where millions of the most vulnerable and malaria-prone populations live in mud-walled huts. Three hundred sixty households were randomly assigned to either the HD4MC (120 households), IRS (120 households) or control group without any wall treatment (120 households). Entomological indices were assessed using pyrethrum spray catching, CDC light traps and human landing catches. The Actellic^®^ 300 CS toxicity on acetylcholinesterase activity among applicators of HD4MC was evaluated using the Test-mate (Model 400) erythrocyte acetylcholinesterase (AChE) test V.2, whereas toxicity in household occupants was monitored clinically. The Actellic^®^ 300 CS level in house dust was analyzed using reversed-phase high-performance liquid chromatography (RP-HPLC). Entomological indices were compared between the three study arms at 1.5, 3 and 6 months post-intervention. HD4MC- and IRS-treated huts had a significantly reduced malaria vector density and feeding rate compared to control huts. There was no significant reduction in acetylcholinesterase activity at 1.5 and 24 h post exposure. Actellic^®^ 300 CS exposure did not result in any serious adverse events among the household occupants. In conclusion, HD4MC was safe and had comparable efficacy to canonical IRS.

## 1. Introduction

Malaria remains an important global development and public health challenge in Africa. An estimated 249 million malaria cases and 608,000 deaths were reported in 2022 with >95% of these cases and deaths occurring in sub-Saharan Africa [1]. High-transmission countries such as Uganda, Nigeria, Mozambique and the Democratic Republic of Congo bore the heaviest burden of malaria and together accounted for almost half of the reported global cases [1]. Control and prevention strategies targeting malaria parasites (chemotherapy) and mosquito vectors (insecticides) have been deployed in affected communities. Chemotherapeutic approaches include the early treatment of malaria cases with artemisinin-based combination therapies (ACTs) and intermittent preventive therapy for prevention (IPTp) in pregnant women. The World Health Organization (WHO) recommends several chemoprophylactic strategies for national malaria control programs [2]. These strategies include seasonal malaria chemoprevention through the administration of monthly doses of antimalarials in children ≥3 months old during peak transmission seasons and perennial malaria chemoprevention [2]; and mass drug administration [3]. There exists a variety of vector control methods aimed at reducing the frequency of contact between the vector and human host and reducing the vector population and capacity by killing or reducing the longevity of adult mosquitoes and larvae. These methods range from insecticidal to non-insecticidal techniques, including larval source management, entomopathogenic microorganisms, insect growth regulators, and endosymbiotic bacteria [4,5]. The mainstream adoption of some of these tools and technologies at scale in endemic populations remains problematic. Current malaria vector control is by indoor residual spraying (IRS) of WHO-approved insecticides and sleeping under insecticide-treated bednets (ITNs). The reported reduction in malaria burden over the last decades is attributed to scale-up of both IRS and ITNs in combination with chemotherapy. However, the success of chemotherapeutic and insecticide-based control and prevention approaches is undermined by the threat of drug and insecticide resistance [6]. Currently, access to the approved malaria vaccines in clinical practice remains limited [7,8,9].

The renewed call for malaria eradication and elimination emphasizes the optimization of existing interventions including vector control. However, both IRS and ITN control strategies have important shortcomings that leave vulnerable children at risk of malaria. Available evidence shows that the distribution, ownership and proper use of ITNs are inefficient, which undermine its effectiveness [10]. IRS is centralized, expensive (requires special equipment, trained personnel, complex logistics), donor-dependent, does not reach remote villages and fails because of community weariness/insufficient uptake [11]. We developed a user-friendly, locally tailored malaria vector control approach called “House Decoration for Malaria Control” (HD4MC) based on a customary hut decoration practice in rural Uganda where millions of the most vulnerable and malaria-prone populations live in mud-walled huts. Mud walls are traditionally decorated by smearing with freely available colored local materials such as black or red soil mixed with cow dung or wood ash. We hypothesized that incorporating a WHO-approved insecticide, Actellic^®^ 300 capsule suspension (Actellic^®^ 300 CS, commonly known by the chemical name pirimiphos-methyl), into soil plaster for smearing hut walls turns the customary hut decoration into a mosquito control tool because the insecticidal mud wall plaster will repel free-flying mosquitoes entering treated huts and kill mosquitoes landing on the smeared walls. Actellic^®^ 300 CS is a wide-spectrum organophosphate insecticide whose effectiveness against malaria mosquito vectors has been validated using IRS [12,13]. Compared against the limitations and shortcomings of IRS and ITNs, HD4MC is simple (needs little training), affordable, more sustainable (requires only water and soil), might have a higher coverage and community acceptance since it is based on a customary hut decoration practice and could be implemented with community participation. HD4MC is also amenable to re-plastering, thereby circumventing the operational difficulties associated with repeat IRS applications. Here, we report preliminary safety and efficacy data on the performance of HD4MC against the established IRS gold standard in a randomized controlled trial in a rural peasant population in northeastern Uganda.

## 2. Materials and Methods

### 2.1. Study Region and Population

Katakwi district is in the northeastern part of Uganda about 343 km from Kampala City. The district lies at 01°54′54.0″ N, 33°57′18.0″ E (Latitude: 1.9150; Longitude: 33.9550) and borders the Amuria, Napak and Kumi districts in the east, north and northwest, respectively. Katakwi District has a total land area of 2428.8 km^2^ (937.8 sq miles) and is made up of eight sub-counties. This study took place in the seven villages of Abwokodia, Acurun, Ongema, Opoyongo, Oleroi, Akworo and Otujai in Abwokodia parish in Usuk sub-county. The typical rainy season in Katakwi is from March to November with marked peaks in April–May and August–October. Malaria transmission is high during the rainy season, peaking in July (72.0 cases/1000 people/month), and lower during the dry season (16.2 cases/1000 people in February) [14]. The district is predominantly inhabited by a Nilo-Hamitic indigenous population whose main occupation is peasant agriculture and small-scale animal farming. The vegetation cover is predominantly savannah grassland interspersed with seasonal and established swamps. The savanna is broken by homesteads of mud-walled, grass-thatched huts.

### 2.2. Study Design

This was a longitudinal cohort study assessing the safety and efficacy of HD4MC in a randomized controlled trial. The study was conducted between May 2017 and December 2018. Villages were randomly assigned to three intervention arms: HD4MC (Acurun and Opoyongo); IRS gold standard (Oleroi and Otujai); and control (Abwokodia, Akworo and Ongema). Control households were not treated with Actellic^®^ 300 CS but received ITNs.

### 2.3. Sample Size Calculations

We assumed that a 50% reduction in mosquito density was achievable based on Presidents Malaria Initiative data from Apac District—an area of comparable malaria endemicity. With a sample size of 140 huts in the HD4MC and 140 huts in the control arm (allowing for 15% drop-out), the study had a 90% power to detect a 50% reduction in mosquito density by the intervention at α = 0.05 (two-tailed). We used a sample size calculator (http://www.stat.ubc.ca/~rollin/stats/ssize/n2.html, accessed on 10 April 2016) based on an assumption of a mean catch of 25 female *Anopheles gambiae* (*An. gambiae*) [15]/trap/night in the control group and a catch of 12.5 or less in either intervention with a standard deviation (SD) of 30. The SD was calculated from the range (R) of catches/hut in Apac District using the formula SD = R/4.

### 2.4. Materials

Actellic^®^ 300 CS (pirimiphos-methyl) was purchased from Syngenta Crop Protection AG., Basel, Switzerland. Long-sleeved coveralls, face masks, sturdy gloves, gumboots and knapsack sprayers were purchased locally. Soil and water were obtained from the villages of intervention.

### 2.5. Study Procedures

#### 2.5.1. Baseline Entomologic Surveys

Baseline entomological survey was conducted in 4 villages (Ongema, Abwokodia, Otujai and Acurun) in August 2017 using pyrethrum spray catching (PSC), Centers for Disease Prevention and Control Light Trap (CDC LT) and Human Landing Catches (HLCs) following standard methods [16]. **PSC:** Briefly, twelve (12) houses inhabited the previous night were randomly selected in each of the 4 villages for the collection of adult mosquitoes using the standard PSC method [16] for seven consecutive days. Prior to PSC, all animals on the veranda, chickens and small furniture were removed from the targeted houses, and white sheets were laid to completely cover the floor and all flat surfaces (under tables as well). With all windows and doors closed, knock down aerosol (Kill it) was sprayed inside and outside the house in clockwise and anti-clockwise directions by skilled entomologists. After 10 min, the white ground sheets were removed from the houses and adult mosquitoes picked using fine forceps. **CDC LT:** Three houses in each of the 4 villages were randomly selected, and a CDC light trap (Model 512; John W. Hock Company, Gainesville, FL, USA) was set 1 m above the floor at the occupant’s bed net from 06:00 p.m. and left overnight until 06:00 a.m. CDC LT activities were conducted for three consecutive days. **HLC:** HLC was carried out during the evening/night for 3 days in the villages of Otujai, Abwokodia and Acurun. Two teams of collectors were placed outdoors and indoors under the supervision of competent entomologists. Indoor collections were carried out from 6:00 p.m. to 6:00 a.m., while the outdoor collections were from 6:00 p.m. to 10:00 p.m., with the assumption that people turn in to sleep in their rooms by 10:00 p.m. and thus are not at risk of outdoor biting after 10:00 p.m. Trained collectors/volunteers exposed their legs to the knee to serve as bait and sat as quietly as possible. Once the collector felt the mosquito landing, the flashlight was turned on to see the mosquito, which was then collected with an aspirator and placed inside a net-covered paper cup. A different paper cup was used for each hour of collection and labeled accordingly. To avoid the limitations of landing collections, which include variations in the attractiveness of human hosts/baits to mosquitoes, collectors switched sites every hour and rotated in batches between indoors and outdoors. The collectors were put on malaria prophylaxis to prevent them from being infected with malaria.

For all procedures, the mosquitoes collected in each house were stored in separate petri dishes that were appropriately labeled (collection date and hour, village, household number). Specimens were transported to the Med Biotech Laboratories field laboratory, and *Anopheles* mosquitoes were morphologically identified according to Gillies and DeMeillon keys [15].

#### 2.5.2. Household Selection and Randomization

All households within each of the seven villages were enumerated and mapped using handheld global positioning system units (Garmin e-Trex 10 GPS unit, Garmin International Inc., Olathe, KS, USA). Using a computerized number generator, every 5th household from each village was approached consecutively, and 120 households were enrolled per intervention arm (HD4MC, IRS and control). A household was defined as any single permanent or semi-permanent dwelling acting as the primary residence for a person or group of people that generally cook and eat together. The households for the entomologic surveys and cohort studies were selected based on the following criteria: (i) houses where people sleep; (ii) at least one house resident 0.5–10 years of age; and (iii) at least one adult resident available for providing informed consent. The exclusion criteria included the following: (i) failure or refusal to consent; (ii) households with pregnant occupiers; (iii) and households having residents with pre-existing allergies. Participants in the IRS and HD4MC arms and smearers were trained to recognize signs of organophosphate toxicity and advised to contact the study physician as soon as possible in case of suspected toxicity.

#### 2.5.3. Hut Wall Treatment by HD4MC

The HD4MC intervention was carried out in the villages of Acurun and Opoyongo. The HD4MC innovation consisted of the following components: (i) soil from home-owner’s garden; (ii) an insecticide (Actellic^®^ 300 CS); and (iii) water (Appendix A, left and right upper panels, respectively). The best soil–which dries without leaving cracks or fissures–was chosen after several dry runs. The components were mixed to achieve a plaster mix of 2 g of insecticide per square meter of wall surface. Actellic^®^ 300 CS is supplied at a concentration of 300 g/liter and 833 mL per bottle. The WHO recommends the use of 1 g of Actellic^®^ 300 CS per square meter of sprayable surface at a nozzle/control flow valve speed of 550 mL per minute and 1.5 bar pressure with the nozzle 45 cm from the surface being sprayed for IRS [17].

For hand smearing application, we used a final concentration of 2 g of insecticide per square meter of wall surface. The surface area of the wall was calculated using the formula 2 × π × R × H, where π is 3.143, R is the radius of the circular hut in meters, and H is the height in meters (invariably all village huts were circular, Appendix A, bottom panel). A typical village hut with a diameter of 9 m and a wall height of 1.5 m has a surface area of 2 × 3.143 × 9/2 × 1.5 = 42.4 square meters. To achieve 2 g of insecticide/square meter of smeared surface, 85 g of insecticide is required for the hut. About 283 mL (an equivalent of 85 g) of the insecticide was measured out using a measuring cylinder into a basin. The measuring cylinder was rinsed three times with 500 mL of water (1500 mL total), and the rinses were added to the insecticide mix in the basin (Appendix A, top left panel). After mixing to ensure that the milky solution was thoroughly mixed, more water was added to bring the mix to 10,000 mL (Appendix A, top right panel). Four 2 L plastic buckets (8 L capacity) of loam soil from the garden or grounds near the home were measured into a wide plastic basin (Appendix A, bottom left panel) and mixed with the insecticide (Appendix A, bottom right panel). The insecticide mix was stirred gently with a gloved hand into the soil until a plaster mix of the right consistency was achieved (Appendix A, left panel). The plaster mix was then smeared onto the inside walls of the huts with gloved hands in circular motions by trained personnel (referred to as smearers). The smearers donned personal protective equipment that included coveralls, face masks, sturdy gloves and gumboots (Appendix A, right panel). All female smearers received a β-HCG urine test to rule out pregnancy due to concerns about possible health/exposure risks to the unborn child. After 2–3 h, the smeared walls dried, and the occupants returned their properties into the hut. All used materials and waste generated were treated as described in the environmental compliance section.

#### 2.5.4. Hut Wall Treatment by IRS

Indoor residual spraying (IRS) was conducted by trained and experienced entomologists comprising the district vector control officer and personnel from the Vector Control Division, Ministry of Health-Uganda. The necessary planning, procurement and training was completed before the actual house spraying. The households were sprayed using 1 g/m^2^ of Actellic^®^ 300 CS using WHO standard procedures with minor adaptations [17]. Spraying started in the innermost part of the house and worked outwards. IRS was applied in the villages of Oleroi and Otujai.

#### 2.5.5. Control Huts with No Wall Treatment

The villages of Abwokodia, Akworo and Ongema were negative controls, and 120 huts in these villages did not receive any intervention. Hut occupants were supplied with insecticide-treated bednets.

#### 2.5.6. Assessment of HD4MC and IRS Efficacy

The impacts of the interventions on the mosquito density and feeding rate were evaluated using PSC to collect and count wild *Anopheles* mosquitoes caught from control huts, HD4MC- and IRS-treated huts. PSC was undertaken as described in the baseline entomologic survey section. Twelve huts were randomly chosen per treatment arm. The surveys were carried out in October 2017 (1.5 months post-intervention), December 2017 (3.0 months post-intervention) and March 2018 (6.0 months post-intervention).

#### 2.5.7. Assessment of Residual Insecticidal Activity on Treated Walls

The post-intervention residual insecticidal activity of HD4MC- and IRS-treated hut walls were carried out using WHO cone bioassays [18,19] in October (1.5 months post-intervention), December 2017 (3 months post-intervention) and March 2018 (6 months post-intervention). We employed 2–5-day-old susceptible adult female *An. gambiae sensu stricto* (*An. gambiae ss*) Kisumu strain mosquitoes that were reared at the Vector Control Division-MoH insectary. Three huts were randomly selected from each of the treatment arms (control, HD4MC and IRS). Three cones were mounted at different heights (upper level (1.5 m), middle level (1.0 m) and lower level (0.5 m)) on the wall surfaces for each hut. Ten (10) female *An. gambiae ss* Kisumu strain were introduced in each of the cones and exposed for 60 min. The knock down time for the female *An. gambiae ss* Kisumu strain was recorded at 0, 10, 20, 30,40, 50 and 60 min. After 60 min, the exposed mosquitoes were transferred into paper cups where they were fed with 10% glucose solution and monitored up to 24 h. The knock down rate of mosquitoes was determined by recording the number of mosquitoes lying down every 10 min for the first one hour, and the final mortality rate was recorded at the end of the twenty-four-hour holding period.

#### 2.5.8. Environmental Safety Compliance

Reusable wears were washed and air dried. All solid waste was swept off the floor, collected and disposed into a pit latrine or a hole dug in the garden in the absence of a latrine. Liquid waste was disposed of in a communal soak pit dug in the village according to strict World Health Organization [20,21] specifications. Used empty bottles were collected, transported and incinerated by a local service provider licensed by the Uganda National Environment Management Authority.

#### 2.5.9. Safety Assessment

The safety of HD4MC and IRS was assessed using different but complementary approaches including the biomonitoring of dust samples from treated huts for organophosphate content, measuring pseudocholinesterase activity levels in blood collected from intervention applicators (smearers and sprayers) and clinical monitoring of subjects living in the treated huts as described below.

Biomonitoring of organophosphate levels in dust samples from treated huts: Actellic^®^ 300 CS, the insecticide employed in HD4MC and IRS, is an organophosphate (OP) known by the chemical name pirimiphos-methyl (PM). Household dust samples were collected from the sweepings of 9 village huts: HD4MC huts (N = 3), IRS huts (N = 3) and control huts (N = 3). These samples were shipped to the laboratory of Dr. Mark Paine (Liverpool School of Tropical Medicine) for OP analysis. Briefly, dust samples (approximately 1 g each) were weighed and transferred to 10 mL glass tubes. One milliliter of acetonitrile, spiked with 100 µg of both pirimiphos-methyl (PM) and dicyclohexyl phthalate (DCP) as internal standards, was added to each tube. The mixture was vigorously vortexed for 2–3 min, and then 0.5 mL aliquots were centrifuged at 15,000 rpm for 15 min to remove debris. Sample extracts and analytical standards were analyzed using reversed-phase high-performance liquid chromatography (RP-HPLC) following the method described in Fuseini et al. (2020) [22] with minor modification. The HPLC system was equipped with a Hichrom ACE 5C18 column (250 × 4.6 mm id), and the mobile phase consisted of 90% acetonitrile in water with 0.1% phosphoric acid. UV detection was employed at 232 nm. The injection volume was 10 µL per sample, and the run time was 20 min. Both the PM and DCP concentrations in samples were determined by comparing their respective peak areas to those of the corresponding analytical standards (PESTANAL^®^, analytical standard, Sigma-Aldrich, Gillingham, UK). The values were normalized against the corresponding internal standard (DCP) readings. The calculated PM concentrations were further adjusted based on the recovery of the spiked PM levels (100 µg/mL). The final PM content was expressed in parts per million (ppm, weight/weight), calculated based on the corrected amount detected per 1 g of dust sample.

Measurement of acetylcholinesterase activity in blood: The effect of organophosphate on the AChE activity among applicators of HD4MC was assessed using the Test-mate (Model 400) erythrocyte acetylcholinesterase (AChE) test V.2 (EQM Research, Inc., Cincinnati, OH, USA) as per the manufacturer’s guidelines. The test system is based on the Ellman colorimetric method in which acetylthiocholine is hydrolyzed by AChE, producing carboxylic acid and thiocholine, which reacts with the Ellman reagent (dithionitrobenzoic acid) and turns yellow. The rate of color formation is proportional to the amount of AChE. Measurements were performed in duplicates on samples collected at 3 time points (pre-exposure, and at 1.5 and 24 h post exposure). Classically, a fall in AChE activity by 40% following exposure is considered a toxic/severe adverse event, and one should stop handling the insecticide [23]. The administration of atropine or pralidoxime is recommended for a reduction of >80% below baseline [24]

Clinical monitoring: The occupants of intervention households were closely followed up for severe adverse events or signs of organophosphate poisoning. Severe organophosphate toxicity presents as unresponsiveness, pinpoint pupils, muscle fasciculations and diaphoresis. Additional symptoms can include emesis, diarrhea, excessive salivation, lacrimation and urinary incontinence [25]. Home visits were conducted daily during the first week, weekly during the first month and subsequently monthly.

### 2.6. Statistical Analysis

All statistical analyses were performed using GraphPad Prism (version 10.1.2 for Windows, GraphPad Software, Boston, MA USA. Categorical variables including sex, age and the use of ITNs were summarized and tabulated as proportions and/or percentages and the median [interquartile range (IQR)] for non-normally distributed continuous variables. The number and/or proportion of mosquitoes collected in the different treatment arms were used to assess the efficacy and residual activity of HD4MC and IRS. The mean number of *Anopheles* (density) and number of blood-fed among the three treatment arms were compared using one-way analysis of variance, while the difference in the mean between 2 groups was compared using Student’s *t* test. The difference in mean AChE activity in ‘overall smearers’ and in female smearers only at baseline and 1.5 and 24 h post exposure were compared using paired *t* tests. Baseline/pre-exposure blood samples from ‘smearers’ served as a reference for post-intervention AChE levels in the same subject due to the lack of normal reference ranges and high variability in AChE levels. The difference in the mean AChE activity levels by sex was analyzed using unpaired *t* test. The difference in the median levels of PM in the households treated with HD4MC versus IRS was analyzed using the Mann–Whitney U test. The difference in the proportion of adverse events between HD4MC and IRS was analyzed using the chi square test.

## 3. Results

### 3.1. Participant Demographics

A total of 360 (120 per treatment arm) randomized households were enrolled in the study. The demographic characteristics of the household inhabitants are shown in Table 1. There was no difference in the participant composition by biological sex or age across the three treatment arms. About 51% of the household inhabitants were female, while 29% were children under 5 years old (Table 1). Insecticide-treated bednets were the only vector control tool used in the study households. The proportion of participants using insecticide-treated bednets was higher in the ‘control’ (47%) compared to the HD4MC- (24.7%) or IRS (28.3%)-treated groups (Table 1).

### 3.2. Baseline Malaria Vector Composition, Density and Distribution

We conducted a baseline survey to characterize the malaria vector densities, species composition, biting and resting behavior and infectivity in the study area prior to the intervention. A total of one thousand seventy-seven (1077) malaria vectors/mosquitoes were collected using three different entomological techniques. The techniques employed served as proxies for investigating different vectoral behaviors (PSC = feeding and resting behavior, CDC LT = indoor feeding behavior, HLC = human seeking and biting behavior). About 20% (219/1077), 23% (247/1077) and 57% (611/1077) of the mosquitoes were collected using PSC, CDC LT and HLC, respectively (Table 2). Out of 1,077 total vectors collected, 76% (823/1077) belonged to the *Anopheles gambiae sensu lato* (*An. gambiae s.l*), while 24% (254/1077) were *Anopheles funestus sensu lato* (*An. funestus s.l*) [15]. Malaria vectors exhibited a spatial distribution in the study area, as no *An. funestus s.l* mosquitoes were caught using the PSC and CDC LT methods in the villages of Ongema and Otujai, and only four were caught using the HLC method in the village of Otujai (HLC was not conducted in Ongema village due to logistical/operational issues). Of the 254 *An. funestus s.l*, 88% (224/254) were caught in Acurun. Similarly, the majority (42.2%) of the *An.gambiae s.l.* were from Acurun. Vector density was the highest and more than double in the village of Acurun (52.4%) compared to the other three villages (Table 2). The feeding rate was evaluated for all the vectors collected by PSC in the four villages. About 96.8% (212/219) of the mosquitoes collected by PSC from the four villages had taken a blood meal (Appendix A), implying a high rate of vector–human contact. Similarly, 95% (579/611) of mosquitoes collected by HLC were caught indoors These data indicate a high risk of malaria transmission in these villages and possible inadequacy of available treated bednets as a malaria vector control tool. The feeding pattern of species-specific female Anopheles vectors was assessed using indoor and outdoor HLC starting at 6:00 p.m. to 6:00 a.m. The peak indoor feeding time for both *An. gambiae* and *An. funestus* was between 1:00 a.m. and 3:00 a.m. (Appendix A). No *An. funestus* were caught outdoors in the villages of Otujai and Ongema (Table 2).

### 3.3. Impact of HD4MC Intervention on Malaria Vector Density and Feeding Rate

Impact on vector density: Malaria vector density estimated by pyrethrum spray catching (PSC) is used as an indicator for the efficacy of IRS and LLINs [26]. The efficacy of HD4MC intervention was assessed against IRS as the gold standard at 1.5, 3 and 6 months post-intervention. This was done by comparing the proportions of female Anopheles mosquitoes collected by PSC in selected households in the different treatment arms (Figure 1A). During the rainy season in October, 1.5 months after implementation, 72.6% (262/361), 21.3% (77/361) and 6.1% (22/361) female *Anopheles* mosquitoes were collected from huts in the control, HD4MC and IRS arms, respectively (Figure 1A). These data demonstrate that female *Anopheles* malaria vectors were reduced by 71% and 92% by HD4MC and IRS, respectively. At the end of the rainy season in late December, 3 months after implementation, 68.1% (81/119), 21.8% (26/119) and 6.1% (12/119) of female *Anopheles* mosquitoes were collected from huts in the control, HD4MC and IRS arms, respectively (Figure 1A). The intense heat and dry conditions that prevailed at the time appeared to have caused a significant drop in the number of caught mosquitoes even in the control arm. These data demonstrate that female *Anopheles* malaria vectors were reduced by 68% and 85% by HD4MC and IRS, respectively. The proportion of mosquitoes collected from the control, HD4MC and IRS huts were 69.4% (104/150), 15.3% (23/150) and 15.3% (23/150) at six months, respectively. This translates to an 85% reduction in vector density in either the HD4MC- or IRS-treated households. Overall, there was a significant reduction in vector density between the intervention and control huts (*p* = 0.0304). There was no significant difference in vector density between HD4MC and IRS at all three time points post-intervention (*p* = 0.2981).

Impact on vector feeding: To better understand the vectoral transmission potential and the impact of HD4MC on human exposure to malaria vector bites, the proportions of blood-fed vectors collected by PSC in the three experimental arms were compared. At 1.5 months post-intervention, the proportion of fed female Anopheles mosquitoes was 76% (183/241), 16% (39/241) and 8% (19/241) in the control, HD4MC and IRS arms, respectively (Figure 1B). These translate into a 78.7% reduction in blood feeding by HD4MC and 89.6% by IRS compared to control. The number of fed female *Anopheles* mosquitoes was 63, 13 and 4 in the control, HD4MC and IRS arms at 3 months post-intervention, translating to 79% and 94% reductions in blood feeding by HD4MC and IRS, respectively. HD4MC and IRS reduced female *Anopheles* mosquito blood feeding by 60.5% and 53.5% at six months, respectively (Figure 1B). Overall, there was a significant reduction in the vector feeding rate between the intervention huts and untreated huts (*p* = 0.0339) but no difference between the HD4MC- and IRS-treated huts (*p* = 0.1654).

Durability of insecticidal activity on treated walls: Three entomological assessments were carried out to determine the residual insecticidal activity on HD4MC- and IRS-treated walls in October (1.5 months post-intervention), December 2017 (3 months post-intervention) and March 2018 (6 months post-intervention) using WHO cone bioassays. The major objective was to determine the residual mosquito killing ability of treated walls at the three time points after HD4MC and IRS. The target mosquitoes employed were 2–5-day-old susceptible female adult *An. gambiae ss* Kisumu strain reared in an insectary at the Vector Control Division, Ministry of Health-Uganda. At 1.5 months post-intervention, 100% of mosquitoes exposed to HD4MC- and IRS-treated walls were knocked down within 60 min of exposure, and 100% died within 24 h after exposure. The results for all three study time points are shown in Table 3. The 24 h mortality of mosquitoes exposed to HD4MC- and IRS-treated walls after 6 months was 90 and 97.5%, respectively.

### 3.4. Safety of HD4MC Intervention

#### 3.4.1. Organophosphate Levels in the House Dust as a Proxy of House Occupant Exposure

The levels of OP in dust samples collected from households assigned to the three treatment arms are as shown in Figure 2. No organophosphate was detected in dust samples from non-treated households. There was no significant difference in OP levels in HD4MC- and IRS-treated households. The median level of OP in house dust was 27.1 (IQR) and 24.2 (IQR) parts per million (ppm) in the HD4MC- and IRS-treated households, respectively. There is no guideline available to define the maximum residue limit (MRL) for OP in house dust. The Australian MRL in bran is 20 ppm [27]. If we take this value as a guideline and comparator, the PM content of the house dust samples remains within the safe range for both HD4MC and IRS. A maximum residue level (MRL) is the highest level of a pesticide residue that is legally tolerated in or on food or feed when pesticides are applied correctly.

#### 3.4.2. Acetylcholinesterase Activity Levels in HD4MC Applicators

OP insecticide toxicity results from the inhibition of acetylcholinesterase (AChE) activity, causing the accumulation of acetylcholine and overstimulation at cholinergic synapses throughout the body [28]. This results in an ‘acute cholinergic crisis’ with bradycardia, hypotension, coma and acute respiratory failure, which requires immediate medical intervention [27]. The effect of organophosphate on acetyl cholinesterase activity was assessed in smearers before and after application of HD4MC using Student’s *t* test. Although there was a visual decline in the mean AChE activity level at the different time points following exposure (pre-exposure, and at 1.5 and 24 h post-exposure), this was not statistically significant (*p* = 0.2456) (Figure 3A). However, we observed a striking and significant reduction in AChE activity in females compared to males even under pre-exposure conditions (*p* = 0.014) (Figure 3B). In comparison to baseline, the AChE activity level was significantly reduced at 24 h post-exposure among female smearers (*p* = 0.0134) (Figure 3C).

#### 3.4.3. Clinical Adverse Events in House Occupants in Treated Households

Household occupants were closely followed up for adverse events or signs of OP poisoning. An adverse event was defined as any undesirable experience following the application of HD4MC or IRS. The visits were conducted daily during the first week, weekly during the first month and subsequently monthly. Serious adverse events (OP toxicity) were defined as unresponsiveness, pinpoint pupils, muscle fasciculations and diaphoresis. Adverse event symptoms can include emesis, diarrhea/abdominal pain, excessive salivation, lacrimation, headache, body itching and rash and urinary incontinence [25]. Home visits were conducted daily during the first week, weekly during the first month and subsequently monthly. Table 4 shows the proportion of adverse events diagnosed during follow-up. There was no serious adverse event associated with either HD4MC or IRS treatment. However, other adverse events registered were headache (14.7%), itching (13.8%) and body rash (10.1%), which resolved during follow-up. However, a major complaint by 36.7% of house occupants was the strong smell of PM in both the HD4MC and IRS arms. There was no significant difference in the number of adverse events registered in IRS versus HD4MC. The results represent complaints recorded within the first week. No IRS- or HD4MC-associated complaints were registered after a month post-intervention.

## 4. Discussion

Millions of vulnerable populations in malaria-endemic regions live in mud-walled grass-thatched or iron-roofed huts. In this pilot study, we investigated the effect of incorporating a WHO-approved insecticide into soil plaster customarily used to decorate hut mud walls to give them a smoother and pleasant or colorful appearance. We have termed this innovative approach Hut Decoration for Malaria Control (HD4MC). HD4MC had a knock down and killing activity on adult female *Anopheles* mosquitoes raised in an insectary and reduced the population density and feeding capacity of wild type *An. gambiae* and *An. funestus* in treated huts in direct comparisons with the standard IRS used in malaria control. Although IRS had higher efficacy, at some time points, the difference was not significant. The WHO recommends a residual insecticidal activity of ≥80% mortality beyond 6 months following application [18]. The insecticidal residual activity of HD4MC remained high for up to 6 months post-intervention, at which time point it was comparable to that of IRS. HD4MC was generally safe for both occupants of treated huts and applicators (smearers) based on the outcomes of clinical follow-up and biochemical assays for AChE in blood of smearers as well as analytical chemistry for insecticide in house dust. The frequencies of the major complaints of smell and headache and other systemic side effects were comparable between HD4MC and IRS. Pre- and post-exposure AChE levels were comparable even after 45 days post-intervention in smearers indicating the lack of OP toxicity. The unexpectedly lower pre-exposure AChE levels in the blood of females must be explored in a follow-up study, and if confirmed, then female smearers will need closer monitoring for the risk of OP toxicity. The pirimiphos-methyl levels in floor dust from huts treated with HD4MC and IRS were within the acceptable maximum residue limit (MRL) for bran in Australia. Overall, these data suggest that HD4MC could be an effective and safe mosquito vector control tool whose effect lasts at least 6 months without the need to reapply the soil mud plaster.

Other studies have reported the anti-mosquito effect of paint and wall linings containing organophosphate (OP) insecticide. First, combining OP insecticides and an insect growth hormone in wall paint and pyrethroid-treated Long-Lasting Insecticide-Treated Nets (LLINs) resulted in a one year killing efficacy against *Anopheles coluzzii* in Burkina Faso [29]. Second, the combination of insecticide paint on doors and windows with LLINs led to high but short-lasting mosquito mortality rates in Burkina Faso [30]. Third, insecticidal durable wall linings and net wall hangings treated with pirimiphos-methyl in combination with LLINs provided significant protection against an *An. gambiae* population [31]. These studies also demonstrated the selection of insecticide resistance genes by single insecticide interventions [31,32]. Fourth, studies in Cote D’Ivoire and Nigeria reported high residual insecticidal activity of sprayed Actellic^®^ 300 CS in mud walls for several months against *An. gambiae* and *Culex quinquefasciatus* [33,34], which was probably due to the microencapsulation formulation [34]. By contrast, the activity on sprayed clay walls against *An. arabiensis* in the Democratic Republic of Congo rapidly declined [35]. Finally, several studies in Asia reported that insecticide wall painting led to long-term efficacy against sand flies, thereby confirming the versatility and usefulness of insecticidal mud wall lining beyond mosquito control [36,37,38].

The major strength of this pilot study is that we confirmed the safety of HD4MC in house occupants and applicators. However, the study has several shortcomings. First, we did not investigate the correlation between the reduced female *Anopheles* population density and the prevalence or incidence of malaria in house occupants. Second, we did not monitor residual insecticidal activity beyond 6 months, so it is unclear whether the insecticidal activity remained high or decayed. This information is required to inform any future malaria vector control based on HD4MC. Third, we did not assess insecticide resistance in the study region before and after HD4MC. These shortcomings will be addressed in future research.

Our hypothesis was that the incorporation of insecticide into soil wall plaster as part of mud wall decoration in rural Ugandan villages turns this customary practice into a safe and effective malaria mosquito vector control tool. Our pilot study largely confirmed this hypothesis. These results are significant because millions of people in impoverished communities live and sleep in mud-walled huts whose walls are decorated with colored soil. The study provides preliminary evidence that smearing insecticide-treated soil plaster onto mud hut walls could be an alternative mosquito vector control tool along with IRS at the grassroots level where soil is a natural resource that is freely available in gardens. Community participation, after appropriate training in safe handling of insecticides and smearing walls, could result in widespread acceptance and adoption of HD4MC. However, several unanswered questions remain about the effect of soil type and microbial composition on the texture and appearance of smeared walls (smooth versus cracked appearance on drying), the stability of the insecticide and duration of residual insecticidal activity and ultimately the duration of efficacy against mosquitoes and protection against malaria disease. Future research must address these outstanding questions and provide evidence through randomized controlled trials regarding whether HD4MC protects villagers living in smeared huts against malaria and whether under certain conditions it might contribute to the malaria elimination agenda.

## 5. Conclusions

The HD4MC is uniquely adapted for mud-walled brick or wattle-and-mud walls, which cannot be painted. This type of house construction is widespread in impoverished villages in Africa, Asia and South America. In these regions, HD4MC could be directly adapted for vector control for various local vector-borne diseases including malaria, visceral leishmaniasis, dengue, Chagas’ disease and *Tunga penetrans*.

## 6. Patents

No patent is associated with any part of the work in this publication.

## Figures and Tables

**Figure 1 tropicalmed-10-00004-f001:**
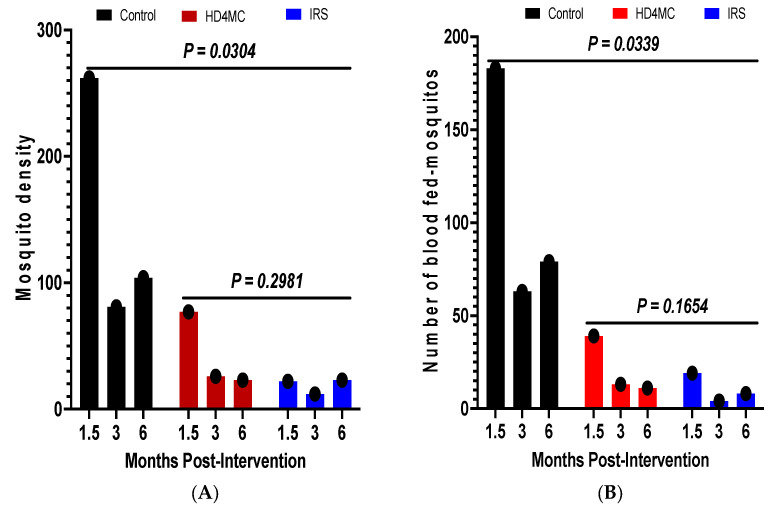
(**A**): Post-intervention vector density. (**B**): Number of blood-fed mosquitoes. (**A**), represents the mean number of mosquitoes collected over a period of 3 days from 12 households in each of the different treatment arms at 1.5, 3 and 6 months post-intervention. (**B**), represents the mean number of blood-fed mosquitoes collected over a period of 3 days from 12 households in each of the different treatment arms at 1.5, 3 and 6 months post-intervention. The difference in the mean mosquito numbers between the 3 groups was analyzed using ANOVA, while the difference in mean mosquito density between HD4MC and IRS was analyzed using Student’s *t* test. HD4MC = house decoration for malaria control, IRS = indoor residual spraying.

**Figure 2 tropicalmed-10-00004-f002:**
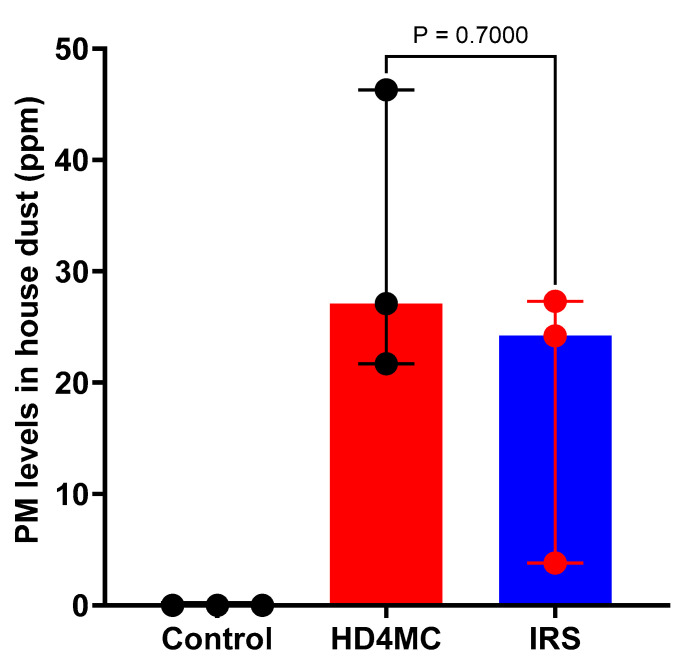
Median (interquartile range–IQR) PM levels in house dust collected from HD4MC- and IRS-treated and control households. Each data value represents a household as well as the mean of two replicate measurements of the PM level in dust obtained from a single household. The difference in the median levels of PM in the households treated with HD4MC versus IRS was analyzed using the Mann–Whitney U test.

**Figure 3 tropicalmed-10-00004-f003:**
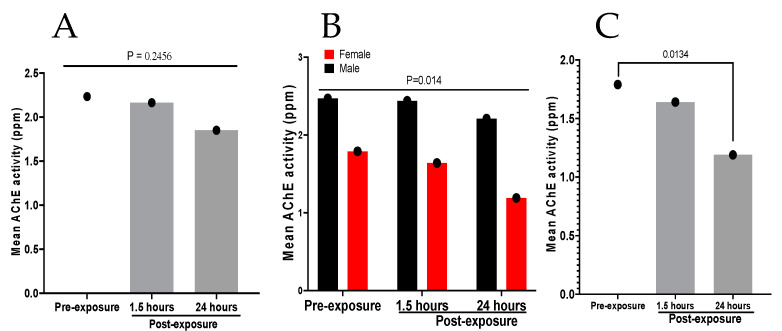
Mean levels of AChE activity in blood obtained from HD4MC applicators at different time points. (**A**), represents the mean level of A*C*hE activity in blood obtained from both male and female (17 persons) smearers before smearing (pre-exposure) and at 1.5 and 24 h after applying HD4MC. The difference in the mean A*C*hE at the 3 time points was analyzed using paired *t* tests. (**B**), represents the mean level of A*C*hE activity in blood obtained from male versus female smearers at pre-exposure and 1.5 and 24 h after smearing. The difference in mean levels between males and females was analyzed using unpaired *t* tests. (**C**), represents the mean level of A*C*hE activity in blood of only female (6 persons) smearers at baseline and at 1.5 and 24 h post-smearing. The difference in the mean A*C*hE activity level was analyzed using paired *t* tests.

**Table 1 tropicalmed-10-00004-t001:** Participant demographic characteristics.

	Intervention			
Participants	Control	HD4MC	IRS	Total
Sex				
Male, n (%)	282 (35.4)	289 (36.3)	226 (28.4)	797
Female, n (%)	304 (35.5)	267 (31.2)	284 (33.2)	855
Age				
<5 years old, n (%)	166 (35.0)	156 (33.0)	151 (32.0)	473
>5 years old, n (%)	395 (34.2)	400 (34.7)	359 (31.1)	1154
Vector control tool, n (%)				
Use ITNs	400 (47.0)	210 (24.7)	241 (28.3)	851

% = percent, n = number, HD4MC = house decoration for malaria control, IRS = indoor residual spraying, sex = biological sex assigned at birth.

**Table 2 tropicalmed-10-00004-t002:** Baseline malaria vector composition, density and distribution.

Village	PSC, n		CDC LT, n		HLC, n		Overall, n (%)	
	AG	AF	AG	AF	AG	AF	AG	AF
Ongema	76	0	62	0	ND	ND	138 (100)	00 (0.0)
Otujai	34	0	67	0	108	4	209 (98.1)	04 (1.9)
Acurun	30	60	60	11	257	153	347 (60.8)	224 (39.2)
Abwokodia	16	3	38	9	75	14	129 (83.2)	26 (16.7)
Total	156	63	227	20	440	171	823 (76.4)	254 (23.6)

ND = Not done; n = number of mosquitoes collected per sub-category; % = percent; AG = *An. gambiae s.l.*; AF = *An. funestus s.l.*; Acurun village = HD4MC, Otujai = IRS, Ongema and Abwokodia = control huts without IRS or HD4MC.

**Table 3 tropicalmed-10-00004-t003:** Knock down and 24 h mortality of lab-reared mosquitoes exposed to treated walls.

Intervention	Residual Insecticidal Activity					
	KD Rate After 60 min			Mortality		
	1.5 Months	3 Months	6 Months	1.5 Months	3 Months	6 Months
Control	0	0	0	0	0	0
HD4MC, % (n)	100 (90/90)	90.0 (81/90)	84.4 (76/90)	100 (90/90)	94.4 (85/90)	90.0 (81/90)
IRS, % (n)	100 (90/90)	83.3 (75/90)	80.0 (72/90)	100 (90/90)	100 (90/90)	98.0 (88/90)

KD rate after 60 min = percent of mosquitoes knocked down after exposure to treated walls for 60 min; mortality = percent of dead mosquitoes 24 h after exposure to treated walls.

**Table 4 tropicalmed-10-00004-t004:** Adverse events.

	Intervention			
Adverse Event	Total (N = 109)	IRS (N = 90)	HD4MC (N = 19)	*p*-Value
Headache, n (%)	16 (14.7)	14 (15.6)	2 (10.5)	0.5735
Itching, n (%)	15 (13.8)	13 (14.4)	2 (10.5)	0.6523
Body rash, n (%)	11 (10.1)	9 (10.0)	2 (10.5)	0.9448
Flu, n (%)	8 (7.3)	6 (6.7)	2 (10.5)	0.5577
Cough, n (%)	7 (6.4)	4 (4.4)	3 (15.8)	0.0668
Abdominal pain, n (%)	2 (1.8)	2 (2.2)	0 (0.0)	0.4903
Fever, n (%)	10 (9.2)	8 (8.9)	2 (10.5)	0.8222
Foul smell, n (%)	40 (36.7)	34 (37.8)	6 (31.6)	0.6104

N = Total population; n = sub-population per category; % = percent. The difference in the proportion of adverse events between HD4MC and IRS was analyzed using the chi square test.

## Data Availability

The raw data supporting the conclusions of this article will be made available by the authors on request.

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
