# Peer review of "Safety and Efficacy of Incorporating Actellic® 300 CS into Soil Wall Plaster for Control of Malaria Vectors in Rural Northeastern Uganda"

_tropicalmed, 2024, doi:10.3390/tropicalmed10010004_

Round 1
Reviewer 1 Report
Comments and Suggestions for Authors
I compliment the authors for embarking on a good study and a difficult field study. However, the design of the study has some deficiencies. The 3 arms are fine but delineating the impact of ITNs which was an intervention in all the arms is not attempted. It would have been better if 2 more control arms namely IRS alone and HD4MC alone could have provided more effective representation for data analysis. Authors may like to reanalyze the available data to address the above issue I have made some suggestions on the MS (attached) to follow for revising the MS.
The flow of the content in the MS was found distorted and the reader is compelled to go back and forth to get the context of the statements in the results and illustrations. Authors may make it more explanatory in the design and methodology sections.

Language and need little refining.
Author Response
Summary
We appreciate and thank all the reviewers for taking time to review this manuscript. We are grateful for the suggestions and constructive comments made to improve the quality and clarity of our write-up. Please find the detailed responses to each comments/suggestion and the corresponding revisions highlighted in track changes in the re-submitted files.
Reviewer 1
Questions for general evaluation |
Reviewers’ evaluation |
Responses and revisions |
Does the introduction provide sufficient background and include all relevant references? |
Yes |
Not applicable |
Is the research design appropriate? |
Can be improved |
We submit that the confounding effects of treated bed nets is evenly distributed between the study groups and is therefore eliminated by the randomization of households. We also agree that the best design would be HD4MC alone, and IRS alone but this was not possible since as a result of a government free bed net distribution program, all households had bed nets. |
Are the methods adequately described? |
Must be improved |
We have restructured sections within the methods and improved on the highlighted statements to enhance clarity |
Are the results clearly presented? |
Must be improved |
We have provided detailed table and figure legends to enhance clarity of results description |
Are the conclusions supported by the results? |
Must be improved |
We believe that the potential confounding effect of ITNs is evenly distributed and therefore, our results support the conclusion that HD4MC is safe and of comparable efficacy and residual activity to IRS. |
Comments and Suggestions for Authors
I compliment the authors for embarking on a good study and a difficult field study. However, the design of the study has some deficiencies.
- The 3 arms are fine but delineating the impact of ITNs which was an intervention in all the arms is not attempted. It would have been better if 2 more control arms namely IRS alone and HD4MC alone could have provided more effective representation for data analysis. Authors may like to reanalyze the available data to address the above issue I have made some suggestions on the MS (attached) to follow for revising the MS.
Response: we are grateful for pointing out a key design strategy to eliminate potential confounding effect of bed net ownership. However, bed net use was evenly distributed between treatment groups, and we feel that this effect was eliminated by the randomization. All households received free bed nets from a government distribution program and the project.
- The flow of the content in the MS was found distorted and the reader is compelled to go back and forth to get the context of the statements in the results and illustrations. Authors may make it more explanatory in the design and methodology sections.
Response: We thank you so much for highlighting the confusion in the flow of statements in the method and results sections. We have revised the methods section by transforming section 2.2 (baseline entomological survey) lines 108-142 on page 3 of 19 to sub-section 2.5.1 lines 165-199 on page 4 of 19. It is now a sub-section of the study procedures. However, Baseline entomological survey results is still presented as Table 2 instead of Table 1 due to the canonical data presentation design beginning with participants characteristics. We also altered and relabeled Figures 2 and 3 to match the methodological description.
Other comments/words/statements highlighted in yellow in the manuscripts that we assumed were either vague and/or had unclear meanings. The reviewer did not give specific reasons for the highlights.
Reviewer highlight: However, both IRS and ITN control strategies have important shortcomings which leave vulnerable children unnecessarily exposed to malaria.
Response: The English has been improved for clarity. The revised statement reads as follows: However, both IRS and ITN control strategies have important shortcomings that leave vulnerable children at risk of malaria (page 2 of 18 lines 65-66).
Reviewer highlight: non-treated control
Response: the study arm labelled “non-treated control” line 114 has been rephrased as “control”. Additional statement has been added as “Control households were not treated with Actellic 300 CS but received ITNs”. The word ‘control’ instead of ‘non-treated control’ has been adopted throughout the manuscript.
Reviewer highlight: Randomization was highlighted in the subheading 2.5.2 ‘Household selection and randomization’.
Response: we assumed that the reason for highlighting the word randomization in section 2.5.2 was probably non-clarity in the paragraph description.
We removed the intervening statement that interrupted the randomization narrative, and the revised statement now reads: “All households within each of the seven villages were enumerated and mapped using handheld global positioning system units (Garmin e-Trex 10 GPS unit, Garmin International Inc., Olathe, KS). Using a computerized random number generator, every 5th household from each village was approached consecutively, and 120 households were enrolled per intervention arm (HD4MC, IRS and no-treatment). A household was defined as any single permanent or semi-permanent dwelling acting as the primary residence for a person or group of people that generally cook and eat together” on page 4 of 18 lines 260-266.
Reviewer highlight: 2 g of insecticide per 189 square meter of wall surface’ on page 4 of 18 lines 285-286.
Response: we could not easily identify any problem with this statement and therefore maintained it. Similarly, we could not figure out the issues highlighted on lines 205 and 224 on page 5 of 18.
Reviewer highlight: HD4MC intervention was assessed against IRS as the gold standard and non-treated huts as a negative control on page 8 0f 18 line 486.
Response: we deleted the phrase non-treated huts as negative control to avoid vagueness.
Reviewer highlight: ‘poor’ on page 13 of 18 line 602
Response: the phrase ‘poor rural villagers’ was rephrased as ‘vulnerable populations’ The new statement now reads as “Millions of vulnerable populations in malaria-endemic regions live in mud-walled grass-thatched or iron-roofed huts”.
Reviewer 2 Report
Comments and Suggestions for Authors
The study investigates the safety and efficacy of incorporating Actellic 300CS into mud wall plaster in rural Uganda for controlling malaria vectors, offering a novel and sustainable approach. I have few comments for the authors to address:
1) For Figures 1A and 1B, please adhere to standard scientific practices. Ensure that each graph presents individual data points from all independent assays, with bars representing the mean values of all experiments. This approach should be consistent with other figures in the manuscript, such as Figures 2C and 3.
2) The current figure and table legends are too brief and lack sufficient details about the experiments. Please expand them to provide concise but thorough descriptions of the experimental setup. Additionally, include the statistical analysis methods used, the number of independent experiments, and the number of replicates for each experiment in the legend.
Others:
Line 148: "wasachievable" should be corrected to "was achievable."
Line 151: "drop out" should be hyphenated to "drop-out."
Line 199: "milky solution was well dispersed" is awkward and should be changed to "the milky solution was thoroughly mixed."
Line 436: "redued" should be corrected to "reduced."
Author Response
Summary
We appreciate and thank you for taking time to review this manuscript. We are grateful for the suggestions and constructive comments made to improve the quality and clarity of our write-up. Please find the detailed responses to each comments/suggestion and the corresponding revisions highlighted in track changes in the re-submitted files.
Questions for general evaluation |
Reviewers’ evaluation |
Responses and revisions |
Does the introduction provide sufficient background and include all relevant references? |
Yes |
Not applicable |
Is the research design appropriate? |
Yes |
Not applicable |
Are the methods adequately described? |
Yes |
Not applicable |
Are the results clearly presented? |
Yes |
Not applicable |
Are the conclusions supported by the results? |
Yes |
Not applicable |
Comments and Suggestions for Authors
The study investigates the safety and efficacy of incorporating Actellic 300CS into mud wall plaster in rural Uganda for controlling malaria vectors, offering a novel and sustainable approach. I have few comments for the authors to address:
- For Figures 1A and 1B, please adhere to standard scientific practices. Ensure that each graph presents individual data points from all independent assays, with bars representing the mean values of all experiments. This approach should be consistent with other figures in the manuscript, such as Figures 2C and 3.
Response: The figure presentations have been redone as suggested. Figures 2A, B and C have been presented as panels instead of individual figures.
- The current figure and table legends are too brief and lack sufficient details about the experiments. Please expand them to provide concise but thorough descriptions of the experimental setup. Additionally, include the statistical analysis methods used, the number of independent experiments, and the number of replicates for each experiment in the legend.
Response: the authors have revised the Table and figure legends as suggested to include statistical methods used for analysis.
Others:
Line 148: "wasachievable" should be corrected to "was achievable."
Response: This has been corrected as suggested.
Line 151: "drop out" should be hyphenated to "drop-out."
Response: This has been corrected as suggested.
Line 199: "milky solution was well dispersed" is awkward and should be changed to "the milky solution was thoroughly mixed."
Response: This has been corrected as suggested.
Line 436: "redued" should be corrected to "reduced."
Response: This has been corrected as suggested.
Reviewer 3 Report
Comments and Suggestions for Authors
1. All species and genus names should be written out in full with the describing author provided the first time they are mentioned. Subsequently, abbreviations should be used throughout the text. For example, see Line 366 ('Anopheles,' etc.).
2. The study mentions that some tests were conducted using human landing catches (HLC = human seeking and biting behavior test). Was ethical committee approval obtained for this?
3. If an abbreviation is to be used in the text, it should be written out fully and clearly at its first occurrence, and the abbreviation should be used consistently thereafter. For example, acetylcholinesterase (AChE).
4. The correct spelling is "acetylcholinesterase," written as a single word. It should be used consistently throughout the text to maintain clarity and accuracy.
5. Pirimiphos-methyl is the active ingredient in the insecticide formulation, while CS refers to the capsule suspension type of the formulation. Therefore, the name should be presented in the text as "Pirimiphos-methyl (Actellic 300 CS)."
6. It has been observed that some words in certain lines are in a larger font size than the standard text. The entire manuscript should be formatted according to the journal's guidelines, ensuring consistent font style and size throughout (e.g., Line 95, Line 153, Line 304).
7. The CDC light trap (Model 512; John W. Hock Company, Gainesville, FL) should be written in full the first time it is mentioned, with abbreviations used thereafter (e.g., Lines 109, 119).
8. The expressions "am" and "pm" should be written separately from the numbers when indicating time. Similarly, terms like "gram (g)" and "milliliter (ml)" should also be written separately from the numbers.
9. A consistent standard should be established for the use of units of time, weight, and volume, such as milliliters, liters, and minutes.
10. In Table 2, the fragmented appearance of species names is not ideal. Instead, abbreviations like 'AG' could be used in place of 'A. gambiae.' The meanings of these abbreviations should be provided in a note below the table. For other species, 'AF' could be used.
11. The expression 'KD60 = percent of mosquitoes knocked down after exposure on a treated wall for 60 minutes' is used incorrectly. KD60 actually refers to the time required for 60% of the population to be knocked down. Therefore, the term 'KD60' should be replaced with 'KD rates after 60 min.'
12. The references within the text and in the reference section need to be formatted according to the journal's style guide (e.g., publication years, punctuation after journal titles, italicization of journal names, correction of abbreviation errors, and proper formatting of species names).
13. The authors have not addressed the results and findings of scientific studies conducted with the same insecticide. I believe that some studies, such as those listed below, should be reviewed and included:
· Fodjo BK, Tchicaya E, Yao LA, Edi C, Ouattara AF, Kouassi LB, Yokoly FN, Benjamin KG. Efficacy of Pirikool® 300 CS used for indoor residual spraying on three different substrates in semi-field experimental conditions. Malar J. 2024 May 15;23(1):148. doi: 10.1186/s12936-024-04912-3. PMID: 38750468.
· Ngwej LM, Mashat EM, Mukeng CK, Mundongo HT, Malonga FK, Kashala JK, Bangs MJ. Variable residual activity of K-Othrine® PolyZone and Actellic® 300 CS in semi-field and natural conditions in the Democratic Republic of the Congo. Malar J. 2021 Aug 30;20(1):358. doi: 10.1186/s12936-021-03892-y. PMID: 34461898.
· Ibrahim KT, Popoola KO, Akure KO. Laboratory Evaluation of Residual Efficacy of Actellic 300 CS (Pirimiphos-Methyl) and K-Othrine WG 250 (Deltamethrin) on Different Indoor Surfaces. Int J Insect Sci. 2017 Nov 2;9:1179543317732989. doi: 10.1177/1179543317732989.
Author Response
We appreciate and thank you for taking time to review this manuscript. We are grateful for the suggestions and constructive comments made to improve the quality and clarity of our write-up. Please find the detailed responses to each comments/suggestion and the corresponding revisions highlighted in track changes in the re-submitted files.
Questions for general evaluation |
Reviewers’ evaluation |
Responses and revisions |
Does the introduction provide sufficient background and include all relevant references? |
Yes |
Not applicable |
Is the research design appropriate? |
Yes |
Not applicable |
Are the methods adequately described? |
Yes |
Not applicable |
Are the results clearly presented? |
Yes |
Not applicable |
Are the conclusions supported by the results? |
Yes |
Not applicable |
Comments and Suggestions for Authors
- All species and genus names should be written out in full with the describing author provided the first time they are mentioned. Subsequently, abbreviations should be used throughout the text. For example, see Line 366 ('Anopheles,' etc.).
Response: this has been corrected as suggested.
- The study mentions that some tests were conducted using human landing catches (HLC = human seeking and biting behavior test). Was ethical committee approval obtained for this?
Response: Ethical approval was obtained from Uganda National Council of Science and Technology and consent was sought from individual participants for the HLC. Proof of ethical approval and consent form have been uploaded for review.
- If an abbreviation is to be used in the text, it should be written out fully and clearly at its first occurrence, and the abbreviation should be used consistently thereafter. For example, acetylcholinesterase (AChE).
Response: This has been corrected throughout the manuscript as suggested.
- The correct spelling is "acetylcholinesterase," written as a single word. It should be used consistently throughout the text to maintain clarity and accuracy.
Response: The spelling of "acetylcholinesterase² has been corrected as suggested.
- Pirimiphos-methyl is the active ingredient in the insecticide formulation, while CS refers to the capsule suspension type of the formulation. Therefore, the name should be presented in the text as "Pirimiphos-methyl (Actellic 300 CS).
Response: The suggestion has been incorporated.
- It has been observed that some words in certain lines are in a larger font size than the standard text. The entire manuscript should be formatted according to the journal's guidelines, ensuring consistent font style and size throughout (e.g., Line 95, Line 153, Line 304).
Response: The manuscript has been formatted as per the journal guidelines.
- The CDC light trap (Model 512; John W. Hock Company, Gainesville, FL) should be written in full the first time it is mentioned, with abbreviations used thereafter (e.g., Lines 109, 119).
Response: The CDC LT is written in full at first mention before the sentence identified by the reviewer.
- The expressions "am" and "pm" should be written separately from the numbers when indicating time. Similarly, terms like "gram (g)" and "milliliter (ml)" should also be written separately from the numbers.
Response: This has been corrected as suggested.
- A consistent standard should be established for the use of units of time, weight, and volume, such as milliliters, liters, and minutes.
Response: This has been corrected as suggested.
- In Table 2, the fragmented appearance of species names is not ideal. Instead, abbreviations like 'AG' could be used in place of 'A. gambiae.' The meanings of these abbreviations should be provided in a note below the table. For other species, 'AF' could be used.
Response: This has been corrected as suggested.
- The expression 'KD60 = percent of mosquitoes knocked down after exposure on a treated wall for 60 minutes' is used incorrectly. KD60 actually refers to the time required for 60% of the population to be knocked down. Therefore, the term 'KD60' should be replaced with 'KD rates after 60 min.'
Response: This has been corrected.
- The references within the text and in the reference section need to be formatted according to the journal's style guide (e.g., publication years, punctuation after journal titles, italicization of journal names, correction of abbreviation errors, and proper formatting of species names).
Response: This has been corrected as suggested.
- The authors have not addressed the results and findings of scientific studies conducted with the same insecticide. I believe that some studies, such as those listed below, should be reviewed and included:
Fodjo BK, Tchicaya E, Yao LA, Edi C, Ouattara AF, Kouassi LB, Yokoly FN, Benjamin KG. Efficacy of Pirikool® 300 CS used for indoor residual spraying on three different substrates in semi-field experimental conditions. Malar J. 2024 May 15;23(1):148. doi: 10.1186/s12936-024-04912-3. PMID: 38750468.
Ngwej LM, Mashat EM, Mukeng CK, Mundongo HT, Malonga FK, Kashala JK, Bangs MJ. Variable residual activity of K-Othrine® PolyZone and Actellic® 300 CS in semi-field and natural conditions in the Democratic Republic of the Congo. Malar J. 2021 Aug 30;20(1):358. doi: 10.1186/s12936-021-03892-y. PMID: 34461898.
Ibrahim KT, Popoola KO, Akure KO. Laboratory Evaluation of Residual Efficacy of Actellic 300 CS (Pirimiphos-Methyl) and K-Othrine WG 250 (Deltamethrin) on Different Indoor Surfaces. Int J Insect Sci. 2017 Nov 2;9:1179543317732989. doi: 10.1177/1179543317732989.
Response: the discussion has been revised to include similar studies on treated mud or clay surfaces conducted by Ibrahim et al 2017, Nwej et al, 2021 and Fodjo et al, 2024, as below:
Fourth, and like our study, studies in Cote D'Ivoire and Nigeria reported high residual insecticidal activity of sprayed Actellic 300 CS in mud walls for several months against An. gambiae and Culex quinquefasciatus [34] which was probably due to the microencapsulation formulation [35]. By contrast, the activity on sprayed clay walls against An. arabiensis in the Democratic Republic of Congo rapidly declined [36].
Round 2
Reviewer 1 Report
Comments and Suggestions for Authors
The authors have given a response to the comments and is acceptable. A few comments marked in the MS have not been responded to. I have given below the comments which can be responded and if needed, make amendments as deemed appropriate.
1. Line 189 - Plesae explain why 2g/Sq.m dose was applied against the WHO recomended dose of 1g/Sq.m
2. Line 205 - How much plaster material was used per hut? This is for information for other investigators.
3. Line 224 - Bednets - Are these LLINSs
4. Table 2 - Indicate the study arm for each village
5. Table 4 - Why the sample size is disproportionate and data of Control arm also could have been added to the table
6. Discussion - The discussion is too verbose and hypothesizes the outcomes discussed without appropriate evidence,. It can be made crisp and relevant to the study outcome.
Comments on the Quality of English Language
Nothing Specific
Author Response
Summary
We are once again grateful for the constructive suggestions and comments made to improve the quality and clarity of our write-up. Please find the detailed responses to each comments / suggestion and the corresponding revisions highlighted in track changes in the re-submitted files.
- Line 189 - Please explain why 2g/Sq.m dose was applied against the WHO recommended dose of 1g/Sq.m
Response: We are aware of the WHO recommendation for indoor residual spraying (IRS). However, we submit that the WHO recommendation of 1 g of ActellicÒ 300 CS per square meter of sprayable surface is based on the results of numerous field investigations of IRS based on a specific nozzle pressure using sprayers. The use of 2 g of ActellicÒ 300 CS per square meter in HD4MC was exploratory due to lack of precedent studies that used the smearing method of application without pressure.
- Line 205 - How much plaster material was used per hut? This is for information for other investigators.
Response: The HD4MC study was conducted in real field, and not simulated semi-field conditions. As such, the huts in the study area and subsequently, the quantity of plaster used for smearing varied across the homesteads depending on the hut size. Nonetheless, the details of materials used and how to calculate quantities per hut is explained in section 2.5.3 Hut wall treatment on page 8-9 lines 172-200.
- Line 224 - Bednets - Are these LLINSs
Response: Yes, the bednets distributed by the national malaria control program and purchased by the study are LLINs.
- Table 2 - Indicate the study arm for each village
Response: Table 2 shows results from an independent baseline survey which was conducted to characterize the malaria vector densities, species composition, and distribution behavior in the study area prior to the intervention. The sample frame used is in no way linked to randomization of the interventions. However, the following information has been included as table legend. Acurun village = HD4MC, Otujai = IRS, Ongema and Abwokodia = control huts without IRS or HD4MC.
- Table 4 - Why the sample size is disproportionate, and data of Control arm also could have been added to the table.
Response: The “N” in table 4 indicates the number of adverse events reported and as such, the numbers differ by treatment arms-these adverse events have been further disaggregated. A total of 109 adverse events were documented, 90 in the IRS group and 19 in the HD4MC arm. We could not compare HD4MC intervention to controls due to availability of an approved/standard vector control tool (IRS). Comparison with controls would make the most sense in the absence of an effective intervention.
- Discussion - The discussion is too verbose and hypothesizes the outcomes discussed without appropriate evidence, It can be made crisp and relevant to the study outcome.
Response: the discussion section has been overhauled